# Advances in Biological Control and Resistance Genes of Brassicaceae Clubroot Disease-The Study Case of China

**DOI:** 10.3390/ijms24010785

**Published:** 2023-01-02

**Authors:** Chaoying Zhang, Chunyu Du, Yuwei Li, Huiying Wang, Chunyu Zhang, Peng Chen

**Affiliations:** College of Plant Science, Huazhong Agricultural University, Wuhan 430070, China

**Keywords:** clubroot disease, *Plasmodiophora brassicae*, R gene

## Abstract

Clubroot disease is a soil-borne disease caused by *Plasmodiophora brassicae*. It occurs in cruciferous crops exclusively, and causes serious damage to the economic value of cruciferous crops worldwide. Although different measures have been taken to prevent the spread of clubroot disease, the most fundamental and effective way is to explore and use disease-resistance genes to breed resistant varieties. However, the resistance level of plant hosts is influenced both by environment and pathogen race. In this work, we described clubroot disease in terms of discovery and current distribution, life cycle, and race identification systems; in particular, we summarized recent progress on clubroot control methods and breeding practices for resistant cultivars. With the knowledge of these identified resistance loci and R genes, we discussed feasible strategies for disease-resistance breeding in the future.

## 1. Overview of Clubroot Disease

### 1.1. The Discovery and Distribution of Clubroot Disease

The pathogen causing clubroot disease in crucifer plants is called *Plasmodiophora brassicae*, and belongs to the genus *Plasmodiophora* in the phylum *Protozoa*. Clubroot disease has been found in continental European Brassica plants as early as the 13th century, and some reports suggest that its presence may be traceable to earlier than the Roman times [1]. In 1737, cruciferous clubroot diseases were officially reported in England from the west coast of the Mediterranean and southern Europe [2]. Early Scottish people generally believed that clubroot disease was caused by poor soil quality or unbalanced fertilizers [1]. In 1873, Russian biologist Michael Woronin first identified the clubroot pathogen and named it *Plasmodiophora brassicae* [3]. In the late 1960s, Woronin studied the relationship between the host and the pathogen and described the pathogen’s life cycle and the interaction mode with the host [4]. From the late 19th and early 20th centuries, the clubroot disease was brought to Canada by European settlers [5]. In 2003, clubroot disease was officially reported on *B. rapa* and thereafter it became rapidly spread in Canada [6,7]. The incidence of clubroot disease in Asia started in Japan and has become a serious economic problem in Japan and Korea [8]. A variety of names have been given to clubroot disease from different countries and regions, indicating the diversity in the nature of the disease from both the pathogen side as well as the plant hosts [9]. In China, clubroot disease was first found in Taiwan and Fujian in the early 1920s [10], though in recent years it has expanded both to the north and to the west inland regions during agronomy development [11,12,13]. With the expansion of rapeseed cultivation and especially the modern farming activities, the incidence of clubroot disease has rapidly increased; provinces with the heaviest incidence include Sichuan, Hubei, Yunnan, and Anhui (Figure 1) [14].

### 1.2. Life History of Plasmodiophora brassicae

Clubroot disease occurs through a specialized parasitism of cruciferous plant roots. The causative pathogen is *Plasmodiophora brassicae*, and transmission of the disease occurs mainly through resting spores on crop residues from an infected field. The resting spores of *P. brassicae* can survive the winter and remain viable for a considerable time in a frozen state, thus posing great difficulties for disease control. The infestation of host roots by *P. brassicae* is generally divided into two stages: I. primary infection on root hairs and II. secondary infection in root cortical cells (Figure 2). The duration of these two phases varies slightly depending on the physiological subspecies of *P. brassicae* and the host [15]. A method of fluorescent probe-based confocal microscopy was used to investigate the root infection process of *P. brassicae* on *Arabidopsis* roots (Figure 2) [16]. During the primary infection (0–7 dpi, dpi = days post infection), resting spores germinated and produced primary mobile spores; the encapsulated primary mobile spores pierced the host cell wall to produce mononucleated protozoa in root epidermis (1 dpi). Mononuclear Plasmodium underwent mitosis and produced multinucleated zoospores (1–3 dpi), which was then accompanied by cytoplasmic cleavage to produce mononuclear secondary spores (3–4 dpi). Secondary free-living spores were released into root hairs or epidermal cells for primary infection (4–7 dpi). In root epidermal cells, the union of two haploid mononuclear secondary zoosporangium produces diploid mononuclear conidia (2n zygote), a process featuring an increase in chromatin volume and nucleus size. At 8 dpi, the presence of widespread mononuclear secondary plasmodium in root cortical cells marked the establishment of secondary infection. During 10–15 dpi, secondary plasmodium further developed binucleate, tetranucleate, and multinucleate forms, resulting in further occupation of roots by the pathogen and a dramatic increase in the root volume manifested as swelling symptoms. Upon 24 dpi, mononuclear resting spores could be found in cortical cells, marking the completion of one life cycle of *P. brassicae* (Figure 2C).

By comparison of the infestation process of *P. brassicae* on different host plants, it is now clear that *P. brassicae* host (e.g., rapeseed, cabbage) resistance is mainly determined by the secondary infection stage, i.e., during the cortical infestation, while non-host (e.g., rice, wheat, and barley) resistance acts primarily during the epidermal infestation stage [17]. Previous studies have provided important references for resolving the mechanisms of plant resistance and developing green and efficient prevention and control strategies against clubroot disease [18,19].

### 1.3. Identification of Physiological Races of P. brassicae

During an investigation of the pathogenesis of different hosts, researchers found that *P. brassicae* has a complex physiological race. That is, pathogens isolated from different regions might have different genetic backgrounds, which result in different disease phenotypes on a given host plant. Several taxonomy systems have been established for different *P. brassicae* species in the world, including (1) the Williams identification system, (2) the European clubroot differential (ECD) system, and (3) the Sinitic Clubroot differentiation (SCD) system.

#### 1.3.1. Williams Identification System

The Williams identification system was established in 1965, and it is still widely used (Table 1) [20]. This system uses four Brassica species, *Jersey Queen*, *Badger Shipper*, *Laurentian*, and *Wilhelmsburger*, and the classified *P. brassicae* are divided into 16 physiological races [20]. For example, when *Jersey Queen*, *Laurentian*, and *Wilhelmsburger* are susceptible (+) and *Badger Shipper* is resistant (−), the pathogen would be classified as race 1 in the Williams system. 

#### 1.3.2. European Clubroot Differential System

The European Clubroot Differential (ECD) system was established in 1975 [21]. The system included 15 plant hosts which can be divided into three major groups: (1) ECD01–ECD05 was the *B. rapa* group (AA, 2n = 20). Within this group, ECD01–ECD04 were *rapifera* and ECD05 was *Pekinesis*, and they could be infected by all races and serve as susceptible controls. (2) ECD06–ECD10 were the *Brassica napus* L. (AACC, 2n = 28) group; and (3) ECD11–ECD15, the *Brassica oleracea* (CC, 2n = 18) group. These materials were collected from different regions of Europe by individual research groups, which served as representative host standards to classify different pathogen races. A binary recording calculation was used to dictate the race of pathogen based on the disease level from the three groups of hosts (Table 2). For example, if the first group’s (*B. rapa* group) hosts are all susceptible, the second group’s (*B. napus* group) are all resistant, and if only ECD15 of the third group is susceptible, then according to the binary calculation, 0 + 0 + 0 + 0 + 0 = 0/1 + 2 + 4 + 8 + 16 = 31/0 + 0 + 0 + 0 + 0 + 16 = 16, this pathogen race would be recorded as ECD 0/31/16.

#### 1.3.3. The Sinitic Clubroot Differential (SCD) System

Although the ECD system includes three major types of *Brassica* species, there are still considerable regional differences in both the host and pathogen in Asian countries, such as China, Japan, and Korea. Therefore, a SCD (Sinitic Clubroot Differential) system was developed for pathogen race identification in mainland China (Table 3). Previous studies using the Williams system showed that the dominant *P. brassicae* race in of China was pathotype 4 [22]. However, due to the annual variation in pathogen populations and difficulty in standardization, a single-spore isolation method was developed recently by Zhang et al., who characterized *P. brassicae* strains isolated from nine different locations of Chinese cabbage cultivation field and obtained a total of 281 single-spore strains belonging to 15 disease types according to the Williams system, of which disease type 4 accounted for the largest proportion [23]. Bai et al. tested 42 species of *P. brassicae* on 12 different hosts, and developed the SCD system [24].

A total of 14 physiological races (Pb1–Pb14) were included in SCD system, with Pb1 as the dominant race [24]. Rui et al. improved the single-cell isolation protocol and combined the SCD system with the Williams system on strains collected from 11 provinces in China [25]. In terms of resistance breeding, disease-level phenotyping is critical. However, pathogens present in the field always constitute a population and this population is likely to change from year to year. Therefore, resistance developed towards a major pathotype may disappear upon long-term use. Since the pathogen could not be multiplied in vitro, it is critical to compare the disease phenotype based on single-spore method, i.e., a uniform genotype of the pathogen strain. Therefore, a scientific and efficient taxonomy system is very important, and this system must be in accordance with the regions to monitor changes possibly involved in field population. Different pathotypes of *P. brassicae* identified in the field can also facilitate resistance breeding in laboratory and greenhouse contexts.

### 1.4. Damage of Clubroot Disease

As mentioned above, *P. brassicae* infects the roots of cruciferous crops, and secondary infection results in swollen roots and the loss of root function, which in turn affects the development of the above-ground parts. *Brassica* L. is one of the most important genera in the cruciferous family, which include hundreds of agronomic crops such as rapeseed, Chinese cabbage, cabbage, shepherd’s purse, radish, turnip, and many others. Many of them can be infected, but there are species that carry resistance genes and become immune to the disease. The area for cruciferous crop cultivation is expanding with modern agronomy techniques; additionally, the resting spores of *P. brassicae* can survive in soil for 8–12 years or even longer, letting mutations accumulate and contribute the appearance of new physiological races [26]. Warm temperature might be in favor of disease outburst, although so far no direct evidence has been reported. Modern farming measures also constitute a major reason for the rapid spreading of clubroot disease in recent years.

Since the clubroot disease infection can occur even in seedlings, the earlier the onset time, the more severe the disease outcome. At the beginning of the disease, there is no obvious phenotype on the aboveground parts, but the new leaf growth will be significantly inhibited. During the middle and late stages, the aboveground parts of the host plant will have stunted growth, yellowing and wilting starts to occur at the base of the leaves, and root galls of different sizes, shapes, and locations will be formed [27]. With the disease progression, the root xylem will be destroyed, the leaves and stems will eventually wilt, and the reduction of the absorption of water and nutrients by the roots will cause a great loss of yield and in extreme cases no harvest at all [28].

The size and location of the root galls/tumors is the basis for disease level grading. A four-level grading system of clubroot disease is proposed as follows: level 0-normal root, no tumor; level 1-no tumor on the main root, small tumor on lateral roots and fibrous roots; level 2-medium tumor on the main root, large tumor on some lateral roots; level 3-large tumor on the main root and lateral roots, enlargement of the basal part of stem, stunted plant growth [29,30]. In addition, Hu reported a disease-grading method especially developed for rapeseed using a scale of 1–4, which can better reflect and assess the severity of disease symptoms [31].

Due to the nature of soil-borne disease, temperature, soil pH, and humidity are significant factors affecting the germination of resting spores and therefore disease incidence. Studies have shown that soil temperatures of 18–25 °C, humidity of about 60%, and pH values of 5.4–6.5 are the optimum conditions for spore germination [32]. In accordance with the climate zone, Hubei, Hunan, Yunnan, Anhui, Sichuan, Jilin, Liaoning, northeast China, southwest China, and Shandong are the main sites of clubroot disease in China [33]. The disease affects an area of 3.2–4 million hm^2^ per year in China, accounting for more than one-third of the total cultivation area of cruciferous crops. In an extreme year, the affected area can reach 9 million hm^2^, with an average yield loss of 20%–30%, and in extreme cases a loss of more than 60% in the field. Therefore, clubroot disease has become a critical problem for the rapeseed industry and also a great threat for many vegetables; clubroot disease has demanded great attention during recent years as a bottleneck agricultural problem in China [34].

### 1.5. Control Measures of Clubroot Disease

The control of clubroot disease mainly follows the policy of “prevention-oriented, integrated control”. It emphasizes the important role of agro-ecological control in disease management, while coordinating biological and chemical control techniques to ensure maximum socio-economic and ecological benefits. The current control measures for clubroot disease are listed below:(1)Field management: *P. brassicae* is spatially aggregated in soil, with high incidence at entrances and field margins [35]. The viability and longevity of *P. brassicae* are closely related to soil properties, and it has been shown that alkaline addition in soil can reduce the germination rate of dormant spores, decrease root-hair infection, and inhibit the maturation of sporangia and Zoosporangium [36]. Therefore, increasing soil pH with lime has been often used for the management of small acreage incidence [37,38,39]. In addition, it has been shown that high concentrations of calcium, boron, and magnesium have important effects on soil inoculum density [40]. High concentrations of calcium are involved in the induction of relevant defense compounds and the induction of host cell death by reducing dormant spore germination and sporangial development at the same time [41,42]. High concentrations of boron could slow down the development of *P. brassicae* by inhibiting its growth during primary infection stage [43]. Interestingly, clubroot incidence and severity were found to be affected by the level of total and individual glucosinolates between oilseed rape cultivars [44]. Since different crops have difference impact on agronomic residues on soil after growth seasons, crop rotation measures have also been used to avoid pathogen accumulation in open fields.(2)Chemical control: chemical agents are used to inhibit the germination of resting spores. Pre-disease control is critical for disease prevention; current measures consist of pharmaceutical seed dressing, seedbed disinfection, soil fumigation, joint root irrigation, etc. A few conventional fungicides, such as SDD (sodium dimethyl dithiocarbamate), thiram, carbendazim, fluazinam, cyazofamid, and pentachloronitrobenzene, have shown to be effective. In addition, options are provided for combining different chemicals together to achieve better results, such as 58% metalaxyl mancozeb (1500 times dilution) and 75% chlorothalonil (1000 times dilution) for milder years [45].(3)Biological control: the soil contains a large number of microorganisms; previous studies showed that *Trichoderma* and *Streptomyces* spp. can suppress *P. brassicae* in cauliflower both in greenhouse and in the field [46]. *Heteroconium chaetospira* is effective against the development of clubroot disease in cabbage at low to moderate soil moisture [47]. Application of formulated biocontrol agents including *Bacillus subtilis* and *Gliocladium catenulatum* could significantly reduce the incidence of clubroot in *Brassica napus* L. [48]. The endophytic fungus *Acremonium alternatum* was shown to suppress clubroot disease in cabbage and *Arabidopsis thaliana* [49].(4)Breeding for disease-resistance cultivars: Field management can reduce disease incidence to some extent, but it requires a lot of labor and at the same time does not fundamentally solve the problem. Indeed, researchers have tried to quantify the abundance of clubroot pathogens using the qRT-PCR method, although in most cases the pathogen is present in a mixed population [50]. Comparative bioassays performed in growth chambers showed that resistance was under selection pressure, and the use of clubroot-resistant cultivars is recommended when *P. brassicae* DNA exceeds 1300 genes copies per gram soil [50]. Chemical control can be efficient but is costly and causes environmental pollution. Therefore, breeding for resistant cultivars could protect the plant and environment together from the disease; this is the most fundamental and effective way to prevent disease spreading. Resistant genes (R genes) could be identified from plant materials that are naturally immune. Breeding for disease resistance using R genes is not only very effective, but also in line with sustainable development strategies.

## 2. Plant Immune Pathways and R Genes

### 2.1. Plant Immune Response Pathways

Plants do not have specific immune cells or a somatic adaptive immune system, such as that of mammals. During evolution, plants have developed their own immune systems by recognizing invading pathogens (viruses, bacteria, and fungi) through various receptors on the cell surface as well as inside the cells [51,52,53]. Currently, the plant immune system is constituted by two pathways. (1) The primary immune pathway, also known as the PTI (pattern-triggered immunity) pathway, is activated by cell-surface pattern recognition receptors (PRR), and recognizes invading pathogenic microorganisms by microbe-associated molecular pattern (MAMP) or damage-associated molecular pattern (DAMP). (2) The secondary immune response, also known as the ETI (effector-triggered immunity) pathway, is triggered by effectors released by pathogens; plants in turn can evolve resistance genes (R genes) that recognize the effectors and trigger host-immune responses [54] (Figure 3). The most commonly accepted model for PTI-ETI interaction is the “zigzag” model [55]. According to this model, PTI and ETI are temporally and spatially distinct and mediated by different factors, but they also interact on the molecular level and there are considerably overlap partners downstream of the ETI and PTI pathways. PTI is the front line of plant defense against pathogens and stimulates the basal defense, while ETI is an accelerated and amplified response of PTI and is generally more effective in preventing further transmission [56].

### 2.2. Disease-Resistance (R) Genes and the NBS–LRR Protein Family

Most of the plant disease-resistance genes (R genes) identified so far encode proteins of the NBS–LRR (Nucleotide Binding Site–Leucine Rich Repeat) family, which are also known as NLR proteins as the major type for the plant R gene family. The NLR protein consists of three main components: the variable N-terminal structural domain, the NB (Nucleotide-Binding) structural domain, and the C-terminal conserved LRR (leucine-rich-repeat) structural domain [57,58]. Based on the characteristics of the N-terminal structural domain, NLRs are mainly divided into TNL with TIR (Toll-interleukin-1 receptor) at the N-terminal, and CNL with a CC (coiled-coil) structural domain at the N-terminal. The CNL class R genes were found in both dicotyledonous and monocotyledonous plants and significantly more than the TNL class. However, TNL class R genes were detected only in dicotyledons [59,60,61].

During plant immunity, NLR proteins act as intracellular immune recognition receptors, recognizing effectors released by pathogens and triggering immune responses [62,63,64]. The ways in which plant NLRs are involved in resistance are divided into direct and indirect effects. Typical of the “gene-for-gene” model is the interaction between the flax rust resistance fungal gene *AvrL567* and the L protein [65]. In rice, the Avr-Pita176 protein binds directly to the Pi-ta LRD region to initiate an immune response against rice blast fungus [66]. The TNL family member RPP1 (Recognition of Peronospora parasitica 1) in Arabidopsis directly and specifically recognizes the ATR1 (Arabidopsis thaliana Recognized 1) effector variant produced by the foliar oomycete pathogen *Hyaloperonospora arabidopsidis* (Hpa) to trigger an immune response [67].

However, most of the NLR proteins are bound to other host proteins before recognizing the effector (Table 4). The TNL protein RPS4 (resistance to *Pseudomonas syringae* 4) can interact specifically with the transcriptional activator bHLH84 and they mediate transcriptional regulation downstream of immunity [68]. RPS4 can also act in concert with RRS1 (resistance to *Ralstonia solanacearum* 1) to confer recognition of *Pseudomonas* AvrRps4 and *Ralstonia* PopP2 [69,70]. In AvrRps4-triggered resistance, RPS4 crosstalks with SNC1. While SPRF1 acts as a transcriptional repressor, its mutation activates SNC1 and lead to enhanced resistance [71]. The fact that RPS4 can interact with multiple proteins reflects the structural diversity of the protein, but the underlying mechanism regarding whether there is competition between multiple factors is not yet fully understood.

CNL protein RPM1 confers resistance to *Pseudomonas syringae* by recognizing the *Pseudomonas* effector Avrpm1 (ADP-ribosyltransferase) and AvrB through the phosphorylation of RIN4 during infection [72,73]. RPS2 (RESISTANT TO P. SYRINGAE2) is activated in Arabidopsis (At) RIN4 by the *Pseudomonas syringae* effector AvrRpt2, forming the AvrRpt2–RIN4–RPS2 defense-activation module [74]. CRT1 encodes a protein with ATPase activity and is an important mediator of defense signaling triggered by R proteins such as RPS2 [75]. Activation of the RPS5 protein requires PBS1 cleavage to trigger ADP–ATP exchange [76]. In other dicotyledons, the tobacco mosaic virus-resistant CNL protein NRG1 plays an important role in the recognition of the TNL proteins Roq1 and RPP1 [77]. The CNL protein Rx1 in potato interacts with NbGlK1 to regulate the binding affinity for DNA [78]. In monocotyledonous species, this indirect action-induced immune response is also prevalent. In barley, a series of MLA proteins (including MLA1, MLA6, and MLA10), which belong to CC-type NLRs, interact with RING-type E3 ligases and mediate the resistance to powdery mildew fungi (*Blumeria graminis*) [79,80]. In rice, Pigm genes encode a set of NLRs, including PigmR, which mediates broad-spectrum resistance. Additionally, PIBP1, a CNL protein containing an RNA-recognition structural domain (RRM), can interact with PigmR to accumulate in the nucleus in an NLR-dependent manner and directly bind target genes *OsWAK14* and *OsPAL1* A/T cis-acting elements of DNA to activate defense against rice plague [81].

In addition to this, the process of plant immunization is usually accompanied by a hypersensitivity response (HR) or local programmed cell death (PCD). Therefore, maintaining the homeostasis of plant NLR proteins is critical for balancing between immunity and growth [82,83,84,85]. The abovementioned studies fully demonstrate that NLR proteins have key roles in the disease-resistance pathways of different pathogens, both in monocotyledonous and dicotyledonous plants. By influencing the binding and possible recognition of effectors by NLR proteins and downstream helper NLRs, different circuits of immune response pathways involving phytohormones and transcriptional reprogramming are initiated and motivated, leading to resistance and morphological changes accompanied by with disease progression. In this sense, NLR-like R genes are the most important gene resources for disease-resistance breeding.

**Table 4 ijms-24-00785-t004:** NLR protein resistance in different crops and their intercrossing proteins.

	Host Plant	NLR Class	NLR Protein	NLR-Interacting Protein (Type)	Pathogen	Effector	Reference
Dicot	Arabidopsis	TNL	RPS4	bHLH84 (TF)	*Pseudomonas syringae*	AvrRps4	[68]
				RRS1 (Paired NLRs)	*Pseudomonas syringae* *Ralstonia olanacearum*	AvrRps4, PopP2	[69]
							[70]
			SNC1	SRFR1 (TPR domain)	*Pseudomonas syringae*	AvrRps4	[71]
		CNL	RPS5	PBS1 (RLCK VII family kinase)	*Peronospora* *parasitica*	AvrPphB	[76]
			RPM1	HSP90.2 (Chaperone)	*Pseudomonas syringae*	AvrRpm1,AvrB	[72]
				RIN4 (Unknown)	*Pseudomonas syringae*	AvrRpm1,AvrB	[73]
			RPS2	CPR1 (E3 ligase (F-box))	*Pseudomonas syringae*	---	[84]
				RIN4 (Unknown)	*Pseudomonas syringae*	AvrRpt2	[74]
				CRT1 (ATPase activity)	*Pseudomonas syringae*	AvrRpt2	[75]
				MUSE13 (TRAF)	*Pseudomonas syringae*	AvrRpt2	[83]
	Tobacco	CNL	NRG1	EDS1 (Lipase-like)	*Tobacco mosaic virus*	P50	[77]
	Potato	CNL	Rx	GLK1 (TF)	*Potato virus X*	Coat protein	[78]
Monocot	Barley	CNL	MLA10	WRKY1,WRKY2 (TF)	*Blumeria graminis*	---	[80]
				MYB6 (TF)	*Blumeria graminis*	---	[85]
				MLR1 (E3 ligase (RING))	*Blumeria graminis*	---	[79]
			MLA1	MLR1 (E3 ligase (RING))	*Blumeria graminis*	AvrA1	
			MLA6	MLR1 (E3 ligase (RING))	*Blumeria graminis*	---	
	Rice	CNL	PIBP1	PigmR (TF)	*Magnaporthe oryzae*	---	[81]

## 3. Clubroot Resistance (CR) Genes and Resistance Breeding in *Brassica* Species

*Brassica* spp. have been domesticated and artificially selected over a long period of time from three diploid parents, *Brassica rapa* (AA, 2n = 20), *Brassica nigra* (BB, 2n = 16), and *Brassica oleracea* (CC, 2n = 18). The combination of the three allopolyploid species gives rise to diverse and rich members of the Brassica genus, including three tetraploid groups, *Brassica napus* L. (AACC), *Brassica juncea* (AABB), and *Brassica carinata* (BBCC) [86]. These hundreds of Brassica species share a high degree of genome structure similarity and duplication from ancestors, but also have accumulated mutations, gene loss, and functional divergence, leading to phenotypic differences in disease resistance.

In order to create a new germplasm with disease resistance, a parent with CR genes needs to be identified and crossed with an elite parent with other desirable agronomy traits such as a tight shape, high yield, and abiotic stress tolerance. Wide hybridization has been widely used to innovate CR cultivars using the crossing of species within the Brassica genus. Marker assisted selection (MAS) facilitates to narrow down the relevant genetic and chromosomal regions and pinpoint candidate genes. Traditional and molecular breeding are combined to accelerate the CR breeding process.

### 3.1. Progress on CR Loci Mapping and CR Gene Identification

*B. oleracea*, *Raphanus sativus* L. (RR, 2n = 18), and *B. rapa* ssp. *Rapifera* (AA, 2n = 20) are the main resource materials for clubroot resistance (CR) genes since they are immune to the clubroot pathogen. The most widely used materials for clubroot disease resistance are the ECD series of European turnip, especially ECD01, ECD02, ECD03, and ECD04 [87]. The CR loci of European turnips were mainly distributed on chromosomes from the A genome as quality traits. Some candidate CR genes have been identified by fine mapping and functional validation. Diederichsen and Sacristan artificially synthesized allotetraploids using ECD04 and *Brassica oleracea* var. *capitata Linnaeus*, and identified three CR dominant loci, *Pb-Bn1*, *Pb-Bn2,* and *Pb-Bn3* [88].

CR loci identified so far from the ECD series include *Crr1* (A08), *Crr2* (A01), *Crr3* (A03), *Crr4* (A06), *CRa* (A03), *CRb* (A03), *CRc* (A02), *CRk* (A02), *Rcr1* (A03), *PbBa3.1* (A03), *PbBa3.2* (A03), *PbBa3.3* (A03), and *PbBa8.1* (A08) (Figure 4). As shown in Figure 4, most of the CR loci were present on chromosome A03 of the A genome [89]. *Crr3*, *CRk*, *PbBa3.3, CRd*, and *BraCRc* loci are located on upper part of A03 and referred as the “A03-1 cluster”; *CRa*, *CRb*, *CRbki*, *BraA3P5X/G.CRa/bKato1.1*, *BraA3P5X/G.CRa/bKato1.2*, *Rcr1*, *Rcr2*, *Rcr4* and *BraCRa* are located on the lower arm of chromosome A03 and therefore named as “A03-2 cluster” [90]. Among all the CR loci and CR gene candidates, only *CRa* and *Crr1a* have been successfully cloned and functionally validated [91,92].

However, a single resistant variety cannot maintain stable resistance over a long time. Convergent breeding by aggregating disease-resistance genes from different sources in a single material is expected to improve the broad spectrum and persistence of disease resistance in varieties and become a more practical and effective breeding model [93]. In order to do this, more loci and CR genes need to be identified from different germplasms that are immune to the disease. The nature and relationship between these loci need to be evaluated before their utilization as gene resources for resistance breeding.

In 2010, Kamei et al. crossed Japanese radish (CR donor) with Chinese susceptible radish to construct a mapping population with 18 linkage groups; they used AFLP and SSR markers to identify a region of 554 Mb [94]. Matsumoto et al. obtained pure lines with high resistance by mounting three CR genes (*CRa*, *CRk*, and *CRc*), and demonstrated that disease resistance can be elevated by mounting CR genes [95]. By SNP mapping and RNA sequencing, Huang et al. identified two possible CR genes (*Bra019410* and *Bra019413*) from *Rcr2* loci in cabbage [96]. Recently, *Rsa10003637* and *RSA1005569*/*Rsa10025571* were identified as CR loci from radish; a significant correlation was found between the *Rsa10025569* locus and disease resistance in a BC1F1 population [97].

*Arabidopsis*, a model crop in the Cruciferae family, is a good model for analyzing the resistance mechanisms of *P. brassicae*. The earliest analyses of *Arabidopsis* resistance to clubroot disease were mainly performed with multiple metabolic pathways [98,99,100], and the observation of the natural response of *Arabidopsis* to clubroot disease in various locations [101,102,103]. Currently, in addition to the identification of the gene *RPB1*, located on chromosome 1, involved in clubroot disease resistance [104], Jubault et al. identified four additive QTL loci, *Pb-At5.23*, *Pb-At5.1*, *Pb-At1*, and *Pb-At4*, and all of these alleles for resistance were derived from the parent Bur-0 [105]. In addition to this, the homology of well-defined resistance genes on the *Arabidopsis* genome was used to further design designer markers for fine targeting [106].

In the typical Brassica radish, Kamei et al. identified *Crs1*, and found that the genomic region around *Crs1* and the genomes around *Crr3* on turnip (*B. rapa.*) share a common ancestor [94]. Gan et al. identified five QTL loci, *RsCr1*, *RsCr2*, *RsCr3*, *RsCr4*, and *RsCr5*, associated with clubroot disease resistance, with *RsCr1* being homologous to the well-defined locus *Crr1* [107]. Recently, Gan et al. identified a new locus *RsCr6* on chromosome 8 and screened for possible resistance candidate genes *R120263140* and *R120263070* [108].

The investigation of the resistance mechanism of different crops to clubroot disease can help us further understand the resistance loci as well as provide a solid theoretical basis for breeding against clubroot disease.

### 3.2. Genomic and Molecular Markers Associated with Clubroot Resistance

With the development of sequencing technologies, different omics have been used on combinations of different hosts and pathogens to understand plant–pathogen interactions [109,110]. Yu et al. performed QTL analysis on resistant cultivars to *P. brassicae* and mapped three QTLs on chromosomes A02, A03, and A08; one QTL, *Rcr4* on chromosome A03, was responsible for resistance to pathotypes 2, 3, 5, 6, and 8 in the Williams system [111]. The QTLs on chromosomes A02 and A08 were named *Rcr8* and *Rcr9* respectively, and two TNL genes were identified from genomic regions between *Bra020936* and *Bra020861* around *Rcr9* loci in *B. rapa* [111]. On the other hand, proteome and metabolome studies showed differential expression of proteins in lipid metabolism, plant defense, cell-wall repair, hormone production, and signal transduction in response to *P. brassicae* infestation [112].

Marker-assisted breeding has been extensively used for clubroot-resistance breeding. With more genomic information released, genomic sequence has been used more frequently for the development of new markers. Zheng et al. identified five molecular markers for pathotype identification, which can distinguish race P11 from P4, P7, and P9, and similarly P9 from P4, P7, and P11 [113]. Lei et al. validated the genetic stability of two co-dominant markers *CRaEX04-1* and *CRaEX04-3* associated with the *CRa* gene in cabbage using 57 resistant varieties and two genetic populations [114]. Jiang et al. studied the CR locus found in resistant “Kc84R” and identified *BnERF034* as one of the CR genes on chromosome A03 [115]. Indeed, many CR loci have been reported, including the ones recently identified by Wang et al., for two QTLs on A03 and A08, conferring resistance to pathotypes 3H, 3A, and 3D in turnip [116]. The development of markers and identification of genomic regions responsible for clubroot resistance laid an important foundation for marker-assisted breeding for the generation of resistant cultivars with durable resistance to clubroot disease.

### 3.3. Progress on CR Breeding for Clubroot Disease

In 2015, Gao et al. performed disease phenotyping on twenty germplasm of *Brassica napus* L; they found that Huayouza 9 and Huashuang 3 had strong resistance [117]. Using molecular marker-assisted selection, *PbBa8.1* locus of turnip ECD04 was transferred into the elite *B. napus* variety Huashuang 5 to create clubroot-resistant line H5R, which was immune to most of the race 4 pathogens in China [118]. A dominant CR gene CRd was successfully transferred from the self-incompatible line “85–74” to the conventional varieties “W3” and “Zhong Shuang 11”, resulting in two new germplasms, “W3R” and “Zhong Shuang 11R”, respectively [119]. In 2021, Li Qian et al. successfully developed the first hybrid oilseed rape variety “Huayouza 62R”, by hybrid combination of a sterile line Huayouza 62R with a resistant cabbage donor parent Bing409R [80]. Huayouza 62R contains two disease-resistance loci, *PbBa8.1* and *CRb*, and showed excellent performance on field trial in disease areas [120]. Hou et al. used “Hua Resistant No. 5” as the source material and Ogu CMS (cytoplasmic male sterility) recovery line RF04 as the recipient, and created a kale-type spring oilseed rape immune to race 4 pathogens [121]. In an attempt to test the contribution of different CR genes and feasibility to promote the resistance by multiple CR genes in one germplasm, Nadil performed hybridization between kale type 409R containing *CRa* gene with kale type rape 305R containing *PaBa8.1*, and selected progeny with two CR loci. The plants with two CR loci displayed good additive effect for disease resistance, supporting a valid basis for the gene-mounting strategy for CR breeding [122].

In addition to breeding for clubroot disease resistance in oilseed rape, Sun Chaohui et al. developed an early maturing variety of cabbage “Anxiu” with resistance to clubroot disease in several trials in Shandong [123]. Yang et al. obtained a hybrid F1 of *B. oleracea* × *B. napus rape* carrying both clubroot disease-resistance genes and Ogura CMS-recovery genes through distant crosses and embryo rescue [124]. He et al. developed 14 new disease-resistant cabbage varieties using heterozygous crossbreeding and molecular marker-assisted selection techniques to meet the production needs of Yunnan cabbage [125]. A generation hybrid, “Jingchun CR3”, with resistance to clubroot disease and tolerance to the shoots of cabbage was created by crossing two self-incompatible lines, CR1572 and CR1582, by Yu Yangjun et al. [126].

## 4. Conclusions

Clubroot disease is considered as a “cancer” for *Brassica* species; the fast spreading of the disease as well as the risk of losing resistance over time calls for a deeper understanding of the *Plasmodiophora* pathogen and the host pathways leading to disease resistance. In this review, we covered a basic background of the disease distribution and the pathogen’s nature, with a greater focus on the plant ETI pathways and the roles of NLR proteins as R genes for clubroot disease. We strongly believe that with the identification and isolation of more CR (clubroot resistance) genes, more resistance materials could be developed to provide a better safeguard for the agronomy industry of *Brassica* species.

## Figures and Tables

**Figure 1 ijms-24-00785-f001:**
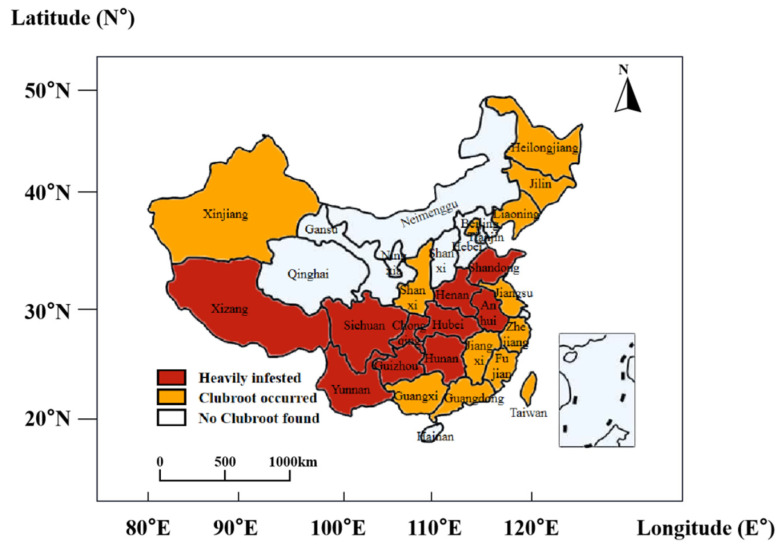
The distribution of clubroot disease in mainland China (Reprinted with permission from Ref. [14]).

**Figure 2 ijms-24-00785-f002:**
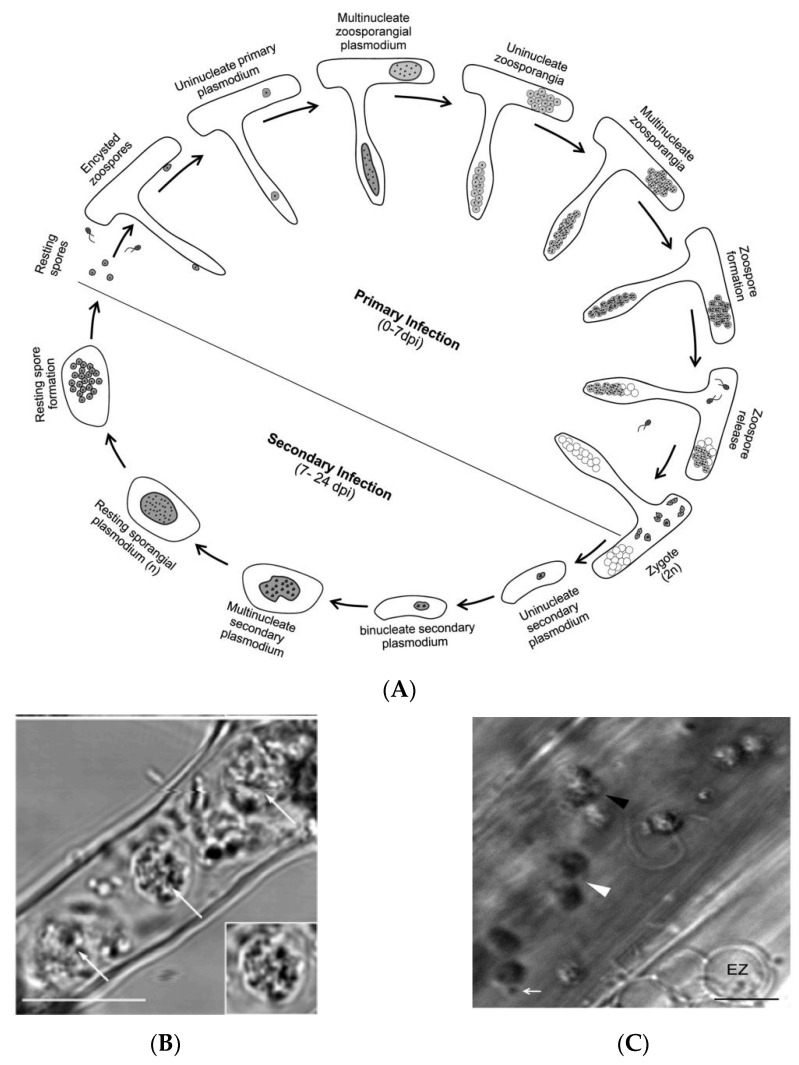
Diagram of the refined life cycle of *Plasmodiophora brassicae* (Reprinted with permission from Ref. [16]). (**A**) Complete life cycle of *Plasmodiophra brassicae*; (**B**) primary zoospore in root epidermis (white arrows); (**C**) secondary zoospore (white arrowhead) and a fusion of two zoospores to form a zygote (black arrowhead).

**Figure 3 ijms-24-00785-f003:**
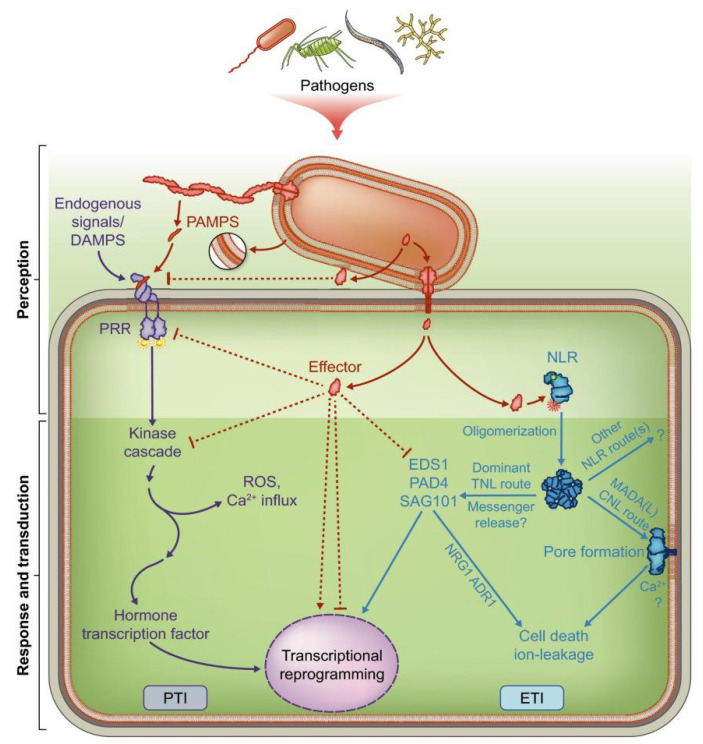
Schematic diagram of the plant immune system (Reprinted with permission from Ref. [3]).

**Figure 4 ijms-24-00785-f004:**
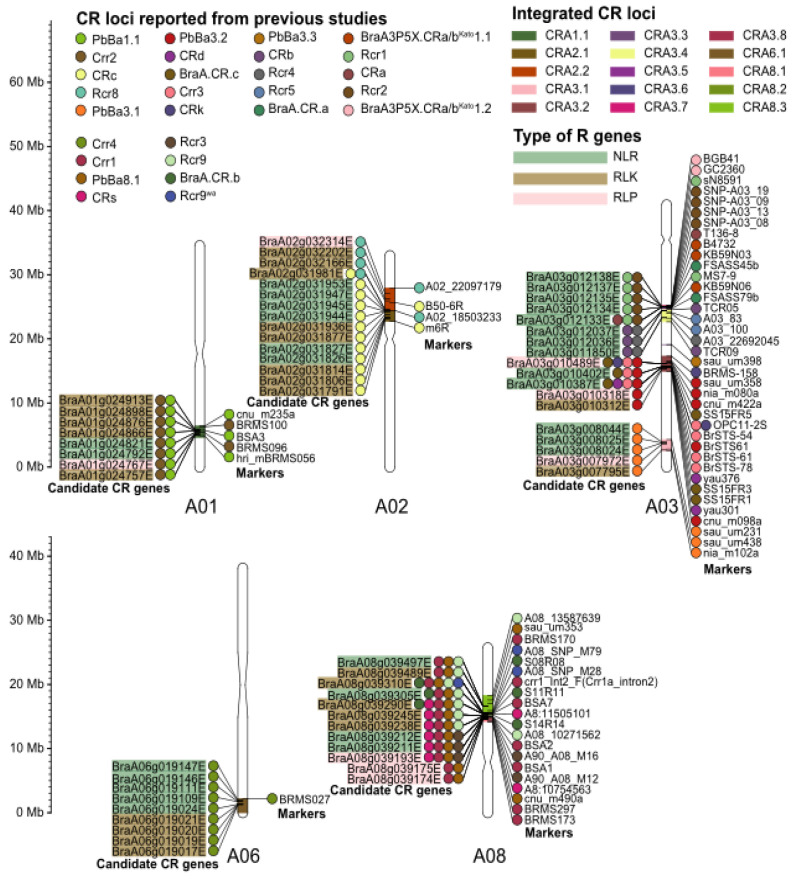
Physical mapping of CR loci (Reprinted with permission from Ref. [89]).

**Table 1 ijms-24-00785-t001:** The Williams classification system [20].

Host Plant	Race															
	1	2	3	4	5	6	7	8	9	10	11	12	13	14	15	16
	+	+	+	+	−	+	+	−	−	+	−	+	−	−	−	−
Jersey Queen																
	−	+	−	+	−	−	+	−	−	+	+	−	+	+	+	−
Badger Shipper																
	+	+	+	+	−	−	−	+	+	−	+	−	+	−	−	−
Laurentian																
	+	−	−	+	−	−	−	−	+	+	+	+	−	+	−	+
Wilhelmsburger																

**Table 2 ijms-24-00785-t002:** The ECD classification system [22].

Differential Number	Differential Host	Binary Numbers	Decimal Numbers
	2n = 20 (*Brassica rapa* L. Sensu lato)		
ECD 01	ssp. *rapifera* line aaBBCC	2^0^	1
ECD 02	ssp. *rapifera* line AAbbCC	2^1^	2
ECD 03	ssp. *rapifera* line AABBcc	2^2^	4
ECD 04	ssp. *rapifera* line AABBCC	2^3^	8
ECD 05	ssp. *Pekinensis* line Granaat	2^4^	16
	2n = 38 (*Brassica napus* L.)		
ECD 06	var. *napus* cv. Nevin line Dc101	2^0^	1
ECD 07	var. *napus* cv. Giant line Dc119	2^1^	2
ECD 08	var. *napus* line Dc128	2^2^	4
ECD 09	var. *napus* cv. Clubrootresistance Dc129	2^3^	8
ECD 10	var. *napus* cv. Wilhelmsburger Dc130	2^4^	16
	2n = 18 (*Brassica oleracea* L.)		
ECD 11	var. *capitata* cv. Badger Shipper	2^0^	1
ECD 12	var. *capitata* cv. Bindsachsener	2^1^	2
ECD 13	var. *capitata* cv. Jersey Queen	2^2^	4
ECD 14	var. *capitata* cv. Septa	2^3^	8
ECD 15	var. *acephala* subvar. Laciniata cv. Verheul	2^4^	16

**Table 3 ijms-24-00785-t003:** The Sinitic clubroot differential (SCD) system [24].

SCD	Pb1	Pb2	Pb3	Pb4	Pb5	Pb6	Pb7	Pb8	Pb9	Pb10	Pb11	Pb12	Pb13	Pb14
WIlliams classification system	2/4/7/11	4	4	4	2/4	4	4	4	4	4	4	4	4	4
H08	−	−	−	−	−	−	−	−	−	−	−	−	−	+
H03	−	−	−	−	−	−	−	−	+	+	+	−	+	+
H01	−	−	−	−	−	−	−	+	−	−	+	−	+	+
H04	−	−	−	−	−	−	+	−	−	+	+	+	+	+
H02	−	−	−	−	+	+	+	−	+	+	−	+	+	−
H05	−	−	−	+	−	+	−	−	+	+	−	+	+	−
H06	−	−	+	−	−	−	−	−	−	−	−	−	−	−
H07	−	+	−	−	−	−	−	−	−	−	−	−	−	−
H12	+	+	+	+	+	+	+	+	+	+	+	+	+	+

**Note:** Pb1–Pb14 are different physiological races of clubroot disease; H01–H08 are different resistant hosts of clubroot disease.

## Data Availability

Not applicable.

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
