# Peer review of "Advances in Biological Control and Resistance Genes of Brassicaceae Clubroot Disease-The Study Case of China"

_ijms, 2023, doi:10.3390/ijms24010785_

Round 1
Reviewer 1 Report
Clubroot disease caused great damage to Brassica species in the world. In this review, the author covered a basic background of the disease distribution and pathogen nature, with more focus on the plant ETI pathways and the roles of NLR proteins as R genes for clubroot, provide a prospect for future work in this field. However, there are small mistakes and some writing in the manuscript could be better.
1. Line149-150: The sentence after “however” could be put into next sentence.
2. Line 176-177: “Many of them can be infected but also there are species that carry resistance genes and become immune to the disease.” “also”should be deleted or put it after “are”.
3. Line 177-179: The sentence after “on one hand” and “on the other hand” has no obvious logical relations.
4. Line 202:Delete apace before “C”
5. 2.1 Immune response pathways in plant
6. In this part, description of R genes and its possible mechanism is too long. Also, you mentioned in this section was just a prospect or what can we do while the R gene of clubroot was obtained. However, your title is Advances in resistance genes of Brassicaceae clubroot disease. Maybe you can change your title to make it consistent with your text.
7. 1.5 When you referred the field management, chemical control of clubroot in your manuscript, proper reference should be added. Additionally, biological control measures could help to against P. brassica in many studies, you haven’t covered it in your manuscript.
8. Line 374: “BrA.CR.a” should be changed into “BraCRa” or “CRa” Please check it all of the text, for example Line 372 “BrA.CR.c”
Reviewer 2 Report
The manuscript “Advances in biological control and resistance genes of Brassicaceae clubroot disease” describes advances made in clubroot disease in Brassicaceae, especially the genetic research and molecular breeding. Overall the article is highly valuable in clubroot disease resistance breeding to enhancement the productivity of Brassicaceae crops.
The article is scientifically and technically sound with valuable objectives, but there are some concerns that need to be addressed.
1. There are many spelling and grammar errors in the manuscript. Please check and correct them carefully. I listed some of them below:
Line 29: use ‘Plasmodiophora brassicae’ for the first appearance in the article, and ‘P. rassicae’ in the following text.
Line 38: the comma after ‘[8]’
Line 42: “include” should be “including”
Line 46: rewrite the sentence ‘Clubroot disease occurs by specialized parasitism of cruciferous plant roots by Plasmodiophora brassicae’
Line 47: the comma after ‘Plasmodiophora brassicae’
Line 52: add ‘and’ before ‘II.’
Line 93: what dose ‘(1)’ mean?
Line 101: the comma
Line 116-138: ‘rapifera’, ‘napus’, ‘napus’, ‘capitata’, ‘acephala’ should be in italics.
Line 175-177: rewrite the sentence ‘Many of them can be infected but also there are species that carry resistance genes and become immune to the disease’
Line 345: change ‘give’ to ‘gives’
Line 347: remove the space in ‘( BBCC)’
2. Line 175: Wild cabbage is not cultivated crop. Pleases change it to cabbage.
3. Line 179: Is ‘Global warming’ a factor increasing the spread of the disease? Pleases provide references.
4. Line 231: Breeding clubroot-resistant cultivars is not biological control, according to its definition.
5. Line 358: Section ‘3.1 Progress on CR loci mapping and CR gene identification’.
All CR loci or genes in Brassicaceae, including Arabidopsis, R. sativus, should be described in this section.
6. Line 395: In section ‘3.2 Progress on CR breeding for clubroot disease’, the authors mainly described the CR breeding in B. napus. It’s better to add descriptions regarding the in progress in other Brassicaceae crops such as B. oleracea, B. rapa.
Reviewer 3 Report
The manuscript “Advances in biological control and resistance genes of Brassicaceae clubroot disease” provides a good review on clubroot distribution, pathotyping and breeding for resistance in China in addition to genetic studies on the disease, plant immune pathways and R genes. However, very little information on biological control has been provided. Therefore, adding more information on bio-control is needed. Furthermore, please consider to make changes in the following:
1. Please carefully check English language each part. Here are two examples: Line 38 page 1, “in recent years it has expand” and Lines 217 &218 page 8, “The current control measures for clubroot disease is listed below”
2. Line 40&41 page 2: Incidence of clubroot in Auhui is very heavy. However, Figure 1 does not support the statement. Please check.
Reviewer 4 Report
I read with very much interest the manuscript entitled ‘Advances in biological control and resistance genes of Brassicaceae clubroot disease’ by Zhang et al. and submitted to IJMS-mdpi.
The manuscript collect very important information, particularly related to China and which leads me to advise authors that the title could be more specific to match with the content, namely and as a suggestion ‘Advances in biological control and resistance genes of Brassicaceae clubroot disease – the study case of China’ would be more adequate, and would reflect the context without taking any value to the manuscript.
On the section 1.5. Control measures of clubroot it is divided into 3 subsections (1, 2 and 3) and next authors start to describe the R genes. In my opinion this should be a continuation of the sub-section 3 from section 1.5. Therefore, to be clear I would advise to authors put 1.5.1. instead of 2. Other way that also could work is to put Control measures as 2. and Plant immune pathways and R as 2.1. and 2.1 Immune response pathways in plants as 2.1.1.
Several important and recent works are missing in this manuscript and to understand better all the pathogen and pathogenicity context of the matter.
Namely, is mentioned in this review regarding the impact of micronutrients in the soil and plant on pathogen infection, something that is already well described in recent literature (eg Botero-Ramirez et al (2021), pathogens mdpi). It is crucial to understand how minerals such as B, Mg and Ca interfere in the pathogen cycle as it seems to delay pathogen primary infections. This should be discussed in the review as it is essential for agronomists and geneticists.
Spacial patterns should also be discussed and compared with what is found in China already.
Same applied to the recent study of Zamani-Noor et al (2021) published in pathogens-mdpi showing the role of Glucosinolate in virulence response of Plasmodiophora brassicae in brassica napus genotypes.
In the discussion of QTL mapping the work of Fengqun et al (2017) on the discovery of six different pathotypes of Plasmodiophora brassicae in Brassica rapa is missing and is worth to be discussed in relation with QTL studies mentioned in the manuscript, particularly if different QTLs found by all studies do share similar genomic positions. This would boost greatly the value of this review.
In a recent work Czubatka-Bie nkowska et al (2021), Pathogens mdpi, also described a formulation to calculate the gene copies specific to P. brassicae that can be used as guidelines, would be interesting to understand how these results may be useful for Chinese farmers and researchers. Please discuss the qRT-PCR methodology in Your review as there are plenty of works that deserve to be discussed in Your review. To this work adds the recent work of Wallenhammar et al (2021).
Authors should also have a working portion regarding existing molecular markers for the topic. The very recent work of Schwelm and Jutta Ludwig-Müller (2021) makes a good resume and is worth to me mentioned and discussed in authors work. This portion goes well with the recent work of Jian et al. (2022), Front. Plant Sci. 13:1014376. doi: 10.3389/fpls.2022.1014376, that recently described novel clubroot disease resistance locus in Brassica napus and related transcripts identifies. This work is very important and should also be included in this manuscript and the results obtained discussed.
Minor comments and for correction
1. Sentences in page 1 lines 24, 27, 32 need the respective references included
2. Authors mentioned ‘ Previous studies provide important references for resolving the mechanisms of plant resistance and developing green and efficient prevention and control strategies against clubroot disease’ – this sentence should be accompanied by all references number corresponding to all studies authors are mentioning.
3. Along the text, authors must uniformize if they are writing in the direct or indirect speech as they mix it.
4. In figure 2. Diagram of the refined life cycle of Plasmodiophora brassicae (cited from: Liu et al, 2020) please substitute cited from to from Liu et al 2020, because it is as such a copy.
5. Regarding the European Clubroot Differential System please give a reference in the text.
6. Line 150 Please give the reference for ‘ Recently, by single-spore isolation method, Zhang et al’
7. A total of 14 physiological races (Pb1-Pb14) was included in the SCD system, with Pb1 as the dominant race [20]. (add the article ‘the’ before SCD system).
8. Chinese cabbage, wild cabbage, shepherd's purse, radish and turnip et al (remove the et al, not sure if authors want to refer ‘… and many others’, or if the reference was deleted by mistake. Which should be included) – line 175
9. Line 178-179: Give references to the following sentence : ‘On one hand, resting spores of P. brassicae can survive in soil for 8-12 years or even longer; on the other hand, the area for cruciferous crops cultivation is expanding with modern agronomy techniques.’
10. Please correct the reference Song et al (2020) Song, W.; Forderer, A.; Yu, D., & Chai, J., Structural biology of plant defence. New Phytol. 2020, 692 - it is 2021 not 2020
11. Figure 3. Schematic diagram of the plant immune system. (cited from: Song et al 2020) should correct the date of publication cited and should substitute cited from to from or source as the figure is a copy of the authors paper figure.
12. Same comment as above and in relation to Figure 4
13. Figure 3 is not mentioned in the text
Round 2
Reviewer 3 Report
It can be accepted for publication.
Author Response
Thank you for your review comments!
Reviewer 4 Report
Point 5: Authors mentioned ‘Previous studies provide important references for resolving the mechanisms of plant resistance and developing green and efficient prevention and control strategies against clubroot disease’ – this sentence should be accompanied by all references number corresponding to all study’s authors are mentioning.
Response 5: Thank you, we have added reference there.
Not provided
Point 9: Line 150 Please give the reference for ‘ Recently, by single-spore isolation method, Zhang et al’
Response 9: Thank you, we have corrected it.
Not correct, authors added the name Zhang et al but no reference (21 concerns to a different work)
